# Effects of Microbial Transfer during Food-Gut-Feces Circulation on the Health of *Bombyx mori*

Lijun Qin,[a,b] Junpeng Qi,[a,b] Guanwang Shen,[a,b] Daoyuan Qin,[a,b] Jinxin Wu,[a,b] Yuwei Song,[a,b] Yang Cao,[a,b] Ping Zhao,[a,b] Qingyou Xia[a,b]

[a]Biological Science Research Center, Southwest University, Chongqing, China
[b]State Key Laboratory of Silkworm Genome Biology, Southwest University, Chongqing, China

Lijun Qin and Junpeng Qi contributed equally to this work. Author order was determined on the basis of seniority.

**ABSTRACT** Change in habitual diet may negatively affect health. The domestic silkworm (*Bombyx mori*) is an economically important oligophagous insect that feeds on mulberry leaves. The growth, development, and immune-disease resistance of silkworms have declined under artificial dietary conditions. In this study, we used *B. mori* as a model insect to explore the relationship between changes in diet and balance of intestinal microbes due to its simpler guts compared with those of mammals. We found that artificial diets reduced the intestinal bacterial diversity in silkworms and resulted in a simple intestinal microbial structure. By analyzing the correlations among food, gut, and fecal microbial diversity, we found that an artificial diet was more easily fermented and enriched the lactic acid bacteria in the gut of the silkworms. This diet caused intestinal acidification and microbial imbalance (dysbiosis). When combined with the artificial diet, *Enterococcus mundtii*, a colonizing opportunistic pathogen, caused dysbiosis and allowed the frequent outbreak of bacterial diseases in the silkworms. This study provides further systematic indicators and technical references for future investigations of the relationship between diet-based environmental changes and intestinal microbial balance.

**IMPORTANCE** The body often appears unwell after habitual dietary changes. The domestic silkworm (*Bombyx mori*) raised on artificial diets is a good model to explore the relationship between dietary changes and the balance of intestinal microbes. In this study, the food-gut-feces microbial model was established, and some potential key genera that could regulate the balance of intestinal microbiota were screened out. Our findings will provide a reference for future research to further our understanding of healthy silkworm development and may even be useful for similar research on other animals.

**KEYWORDS** intestinal microbiome, dysbiosis, artificial diet, *Bombyx mori*, colonizing opportunistic pathogen

Intestinal microorganisms are closely linked to the health of animals and participate in various physiological activities of the host, ranging from growth and development to disease resistance (1). Intestinal microorganisms provide nutrients, participate in the degradation of harmful substances (2), promote growth and development, regulate mating and reproduction, affect the spread of disease (3), participate in immune development, and provide protection against pathogens (4). In addition, the intestinal microbiota of an animal is affected by the external environment, especially its diet (5). For example, feeding a high-fat diet caused a significant reduction in diversity and overall structural shifts in bacterial communities in adult mice, such as a decrease in *Bacillus bifidus* (6). Memory and learning behavior in mice are temporally associated with diet-induced alterations in gut bacteria (7). Under normal conditions, there is a

Address correspondence to Qingyou Xia, xiaqy@swu.edu.cn.

The authors declare no conflict of interest.

range within which the intestinal microbiome can fluctuate; however, drastic environmental changes can cause the intestinal microbiota to become imbalanced. This leads to dysbiosis, which is defined as an imbalance of the microbial communities living in or on the body. Dysbiosis may lead to obesity, metabolic syndrome, and autoimmune diseases (8). A previous study showed that changes in behavior alter the gut microbiota composition, and modifications to the microbiome can induce depressive-like behaviors (9). Therefore, we aimed to explore the relationship between the external environment and intestinal microbial balance.

Compared with mammals, insects have less complex guts and simple diets. Given its high economic value, the silkworm *Bombyx mori* is intensively reared; it is an oligophagous insect that feeds on mulberry leaves, has a short life cycle, and is sensitive to harsh living environments (10). After thousands of years of domestication, the habits, appearance, and cocoons of *B. mori* have changed significantly; however, its feeding habits remain the same (11). These silkworms form an intestinal environment and microbial structure suitable for the digestion of mulberry leaves, which aids in immune-disease resistance and their healthy development (12, 13). However, there are issues associated with the growth cycle of mulberry trees, quality of mulberry leaves in different seasons, and the effects of natural disasters that limit the efficiency of silkworm culture. Therefore, we developed an artificial feed for breeding silkworms to maturity that will overcome these limitations and help with the automation of sericulture. A simple formula for this feed is provided in Materials and Methods. However, the artificial diet considerably alters silkworm rearing methods and environments. This is an issue because silkworms struggle to adapt to changes in their environment, which can affect the filament quality of cocoons, survival rate of young larvae, and resistance to bacterial and viral diseases (10, 14–16). Therefore, we utilized *B. mori* fed the artificial diet (17) to establish a microbial model of the food-gut-environment to study the relationship between the external environment and the balance of intestinal microbes. Furthermore, we aimed to reveal the key factors affecting the healthy development of silkworms fed on this artificial diet and determine how such an artificial diet could lead to a decline in disease resistance and immunity. Our findings will provide a reference for future research to further our understanding of healthy silkworm development and may even be useful for similar research on other animals.

## RESULTS

**Comparison of gut bacteria from *B. mori* fed either mulberry leaves or artificial feed.** There were considerable differences in various characteristics between the silkworms fed mulberry leaves and those fed the artificial diet due to their different diets, including growth and development (weight and instar duration) (see Fig. S1 in the supplemental material) and immune disease resistance, incidence of which was approximately 32.3% at the 5th instar and which resulted in 100% death if no other treatment was applied. Intestinal microbes that were closely related to the diet were chosen as the initial focus of research. Chen et al. (18) showed that the microbial communities are essentially indistinguishable between the foregut, midgut, and hindgut; thus, the whole gut of silkworms (5th instar) fed either mulberry leaves (Mul group) or artificial feed (Art group) was dissected for 16S rRNA gene sequencing.

Species richness and community diversity were estimated by calculating several indexes (e.g., Sobs, bootstrap, and Simpson [Table S1]), but we used the Chao1 and Shannon indexes (19, 20) for visualization. The community diversity of the Mul group was significantly higher than that of the Art group, with more bacterial species (Fig. 1a; other indexes are shown in Table S1a), as demonstrated by "Others" (genera with less than 0.01% combined) in the Mul group, which accounted for 41.16% of the diversity (Fig. 1c). The top three dominant phyla in the two groups were *Proteobacteria*, *Actinobacteria*, and *Firmicutes*, but the proportions were slightly different between the groups (Fig. 1b). At the genus level in the Art group, *Ralstonia* was the most abundant (56.45%), followed by *Rhodococcus* (20.75%), and *Burkholderia-Caballeronia-Paraburkholderia* (9.6%); in contrast, in the Mul group, *Burkholderia-Caballeronia-Paraburkholderia* was the most abundant (23.09%), followed by *Ralstonia* (12.86%) and *Rhodococcus* (4.73%), suggesting that the

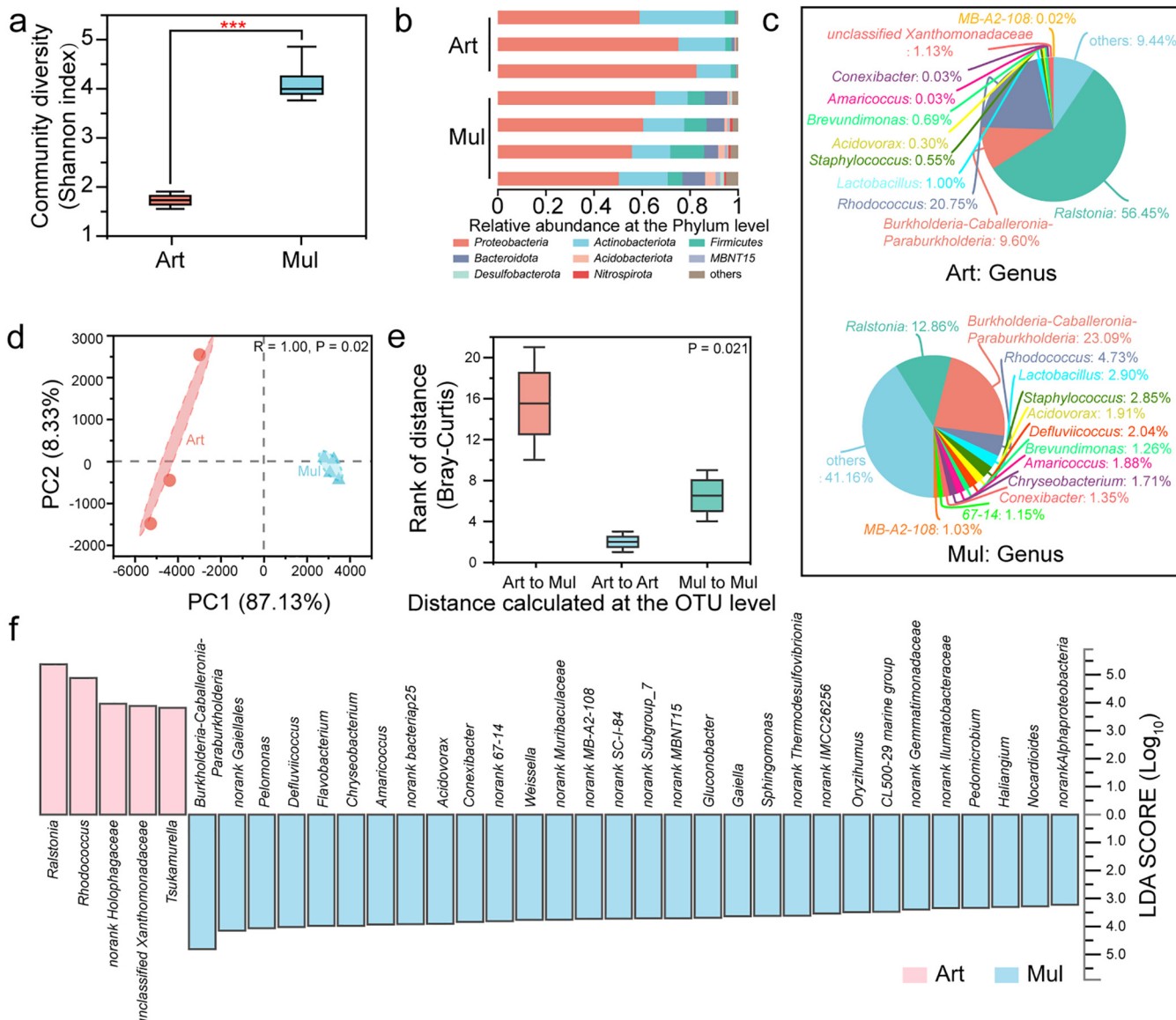

**FIG 1** Intestinal bacterial communities of silkworms fed different diets. (a) Shannon index for community diversity of the test group and variance analysis between groups using Student's *t* test. ***, $P < 0.001$. (b) Relative abundance of bacterial phyla in different samples. (c) Genus-level pie diagram. (d) PCA plot based on the structure of the microbial community at the OTU level for the artificial feed group (Art; red circles) and mulberry leaf group (Mul; blue triangles). The percentage of variation explained by each axis (PC1 and PC2) refers to the fraction of the total variance of the data explained by the constrained factor. Community variation is based on Bray-Curtis distances and an ANOSIM with 999 permutations. (e) ANOSIM for the two groups. The box corresponding to "Art to Mul" represents the distance value of the difference between the groups, and the remaining boxes represent the distance value of the difference within the group; the *y* axis scale represents the size of the distance value. Community variation is based on Bray-Curtis distances with 999 permutations. (f) Species difference analysis based on LEfSe at the genus level. LDA was used to estimate the effect of the abundance of each component (species) on the difference effect. The LDA threshold is 2, and the multigroup comparison strategy is all-against-all (more strict), where only species with differences in multiple groups are considered different species. There were three individuals per sample to average one silkworm.

artificial feed significantly increased the proportion of *Ralstonia*, which became the top genus, whereas *Burkholderia-Caballeronia-Paraburkholderia* significantly decreased, becoming the third dominant genus (Fig. 1c). Principal-component analysis (PCA) (21) showed that the Art and Mul groups were clustered separately at the operational taxonomic unit (OTU) level (Fig. 1d). The difference between the groups was significantly greater than that within the groups (Fig. 1e), where the community structure was significantly different ($P = 0.021$). A total of 35 genera with significant differences were identified using linear discriminant analysis (LDA) effect size (LEfSe) (22). The top three dominant genera (*Ralstonia*, *Rhodococcus*, and *Burkholderia-Caballeronia-*

*Paraburkholderia*) were among these 35 genera, and the LDA score was as high as 5 (Fig. 1f).

There was no significant difference between the Art and Mul groups in the top 20 abundance of KEGG pathway level 3 heat map (PICRUSt2 function prediction) (23), and the top three pathways were "Metabolic pathways," "Biosynthesis of secondary metabolites," and "Microbial metabolism in diverse environments" (Fig. S2a). According to the BugBase phenotype prediction (24), *Ralstonia* highly contributed to "Facultatively anaerobic" and "Stress tolerant," whereas *Burkholderia-Caballeronia-Paraburkholderia* was mainly associated with "Mobile element containing" (Fig. S3).

**Gut microbial signature of diseased silkworms.** During feeding of the silkworms in this study, a bacterial disease with uniform symptoms was observed. At first, the body turned slightly yellow; then the tail decayed, and the body shortened. At the time of death, the abdomen swelled slightly and the whole body was limp (Fig. S4).

Further analysis was performed on the intestines of the diseased silkworms (sick group) fed an artificial diet. We found that the species abundance and community diversity of the diseased silkworms (sick group) were significantly lower than those of the healthy silkworms (healthy group, same as the Art group in Results) (Fig. 2a and b; other indexes are shown in Table S1b). The dominant genera varied significantly between the groups: the dominant *Ralstonia* and *Rhodococcus* in the healthy group were replaced with *Enterococcus* in the sick group, making up over 95%, followed by *Weissella* (Fig. 2c). For this phenomenon, diseased silkworms fed mulberry leaves (Mul_Sick group) were also tested and showed similar symptoms. The dominant genus in the Mul_Sick group was also *Enterococcus* (48.77%), followed by *Lactobacillus* (22.88%) and *Weissella* (10.82%) (Fig. 2e).

In addition, florfenicol, a broad-spectrum veterinary antibacterial agent, was added to the artificial feed. The symbol "(+)" is added to the group name to denote a group that was fed the artificial diet with antibiotics. Species richness and community diversity of healthy silkworms administered antibiotics, the healthy(+) group, were significantly reduced compared to those of the normal healthy group (Fig. 2a and b; other indexes are shown in Table S1b), and the dominant genera also significantly differed. *Weissella* composed approximately 40% and *Enterococcus* composed more than 30% in the healthy(+) group; the microbial composition of the sick(+) group was the same as that for the sick group (Fig. 2c). The Venn diagram showed that the sick(+) group had no unique OTUs, and its OTUs were also found in all healthy silkworms (Fig. 2d). For community structure at the OTU level, all diseased silkworms fed the artificial diet were clustered together. Among the healthy silkworms, there were significant differences between those fed diets of mulberry leaves, artificial feed, and artificial feed with antibiotics (Fig. 2f). The species-related network graph (25) showed that among the top 30 dominant species, *Enterococcus* was negatively correlated with 22 other genera, and there were no positively correlated bacterial genera (Fig. 2g). Therefore, *Enterococcus* proliferation will inevitably inhibit other genera, and other bacteria may also maintain the abundance of *Enterococcus* to a certain extent. *Weissella* and unclassified *Lactobacillales* promoted each other and together antagonized *Halomonas*, *Cutibacterium*, and *Bacillus*.

Next, we counted the number of OTUs of *Enterococcus* (sequences with more than 97% similarity clustered into the same OTU) and the proportion in each group (Table S2). A total of four OTUs were classified as *Enterococcus*, among which OTU856 had a high abundance in each group (mean in each group: Art_Healthy, 43.67; Mul_Healthy, 57.5), which considerably increased in the diseased silkworms (mean in each group: Art_Sick, 12,391.33; Mul_Sick, 4,135.67). We isolated three different strains of *Enterococcus mundtii* from the gut of the diseased silkworms, named CQJ-1, CQJ-2, and CQJ-3, all of which were within OTU856 (Fig. S5a and b). The same symptom phenotype appeared after silkworms were inoculated, the number of *Enterococcus* organisms increased significantly, and the detection of the 16S rRNA gene sequence remained the same (Fig. S5c and d). When silkworms were inoculated with *Staphylococcus sciuri* (isolated independently; named SH), the phenotype of the disease was different, and *S. sciuri* SH led to a small increase in *Enterococcus* (Fig. S5c and d).

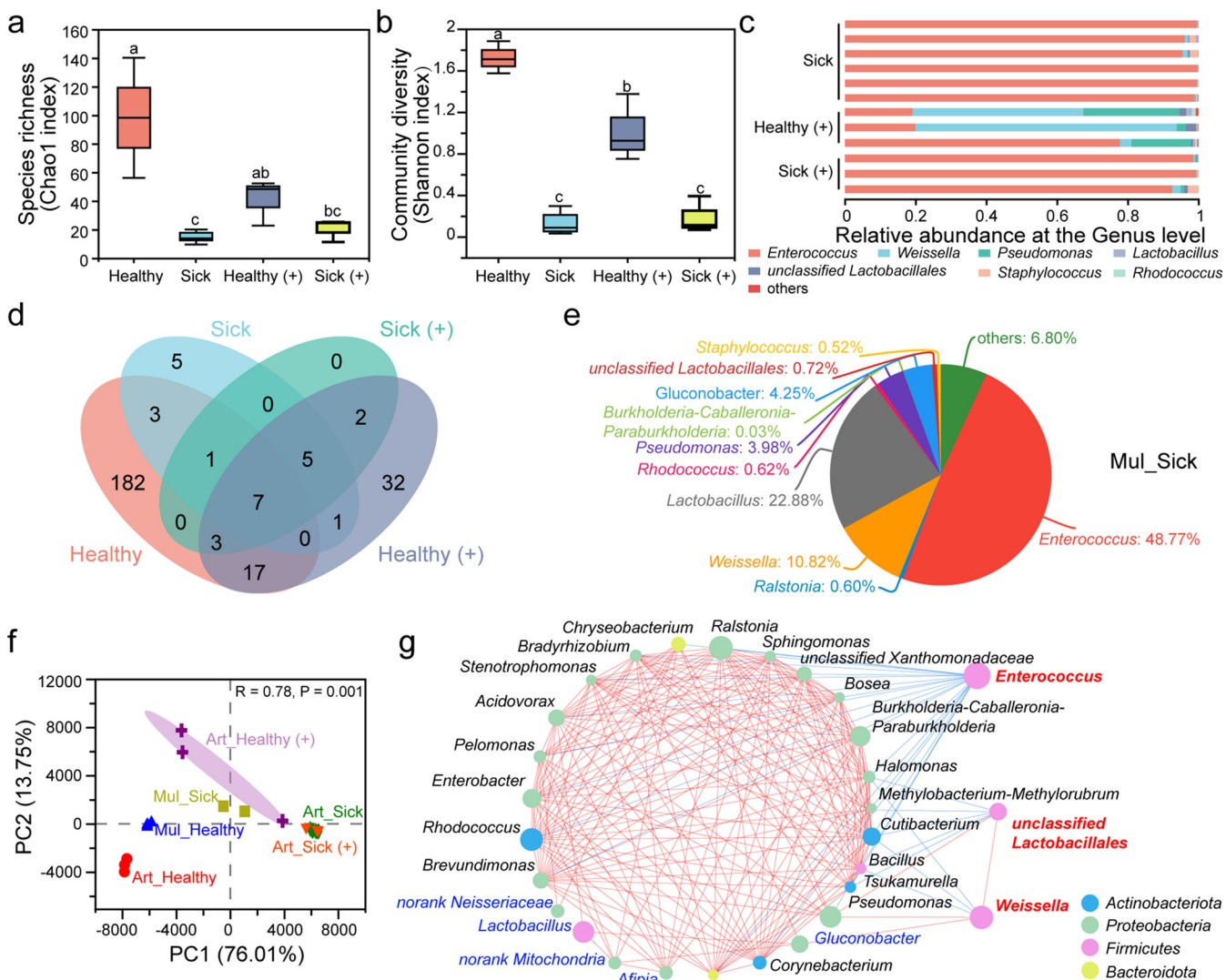

**FIG 2** Effect of antibiotics in artificial feed and bacterial disease. Chao1 index for species richness (a) and Shannon index for community diversity (b) of the test group and variance analysis between groups using the Student's *t* test. Different letters (a, b, and c) above the bars indicate a significant difference at a *P* value of <0.05. (c) Relative abundance at the genus level in different samples. (d) Venn diagram of the three groups at the OTU level. (e) Pie diagram at the genus level of sick silkworms fed on mulberry leaves (Mul_Sick). (f) PCA plot based on the structure of the microbial community at the OTU level. Art_Healthy, gut of the healthy silkworms fed artificial diet without antibiotics (red circles); Mul_Healthy, gut of the healthy silkworms fed a diet of mulberry leaves (blue triangles); Art_Healthy(+), gut of healthy silkworms fed artificial diet with antibiotics (purple crosses); Art_Sick, gut of sick silkworms fed artificial diet without antibiotics (green diamonds); Mul_Sick, gut of sick silkworms fed mulberry leaves (yellow squares); and Art_Sick(+), gut of sick silkworms fed artificial diet with antibiotics (red triangles). The percentage of variation explained by each axis (PC1 and PC2) refers to the fraction of the total variance of the data explained by the constrained factor. Community variation is based on Bray-Curtis distances and an ANOSIM with 999 permutations. (g) Network correlation between intestinal bacteria. The nodes on the left side of the ring are positively correlated at all instances (red lines), and the blue lines indicate negative correlations between the genera. Three individuals per sample were used, to average one silkworm.

Compared with the healthy silkworms fed the artificial diet, the diseased silkworms showed a higher functional abundance of the "Phosphotransferase system (PTS)" and "Starch and sucrose metabolism" pathways (Fig. S2b). Notably, silkworms with healthy phenotypes had a higher proportion of the "Potentially pathogenic" category in the BugBase prediction (Fig. S6a). The antibiotic-free group (healthy group) was mainly contributed to by *Ralstonia*, whereas the antibiotic group [healthy(+) group] was mainly contributed to by *Pseudomonas* (Fig. S6e). However, *Enterococcus* was closely associated only with the "Mobile element containing" category (Fig. S6b).

**Transfer microbiota of food-gut-feces.** After steaming at high temperatures, bacteria are removed from artificial feed but nutrients remain rich; nonetheless, the protection and resistance conferred by mulberry leaf epidermis appears to be lacking (26).

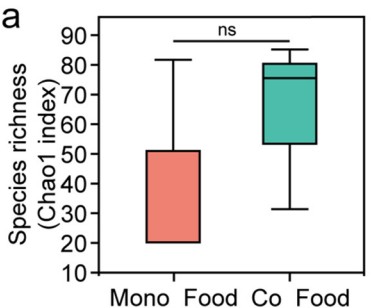
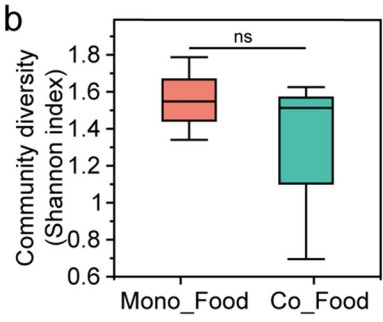
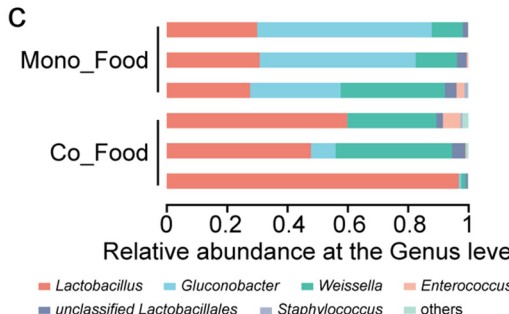

**FIG 3** Microorganisms accumulated in food. Chao1 index for species richness (a) and Shannon index for community diversity (b) of the test group and variance analysis between groups using Student's *t* test. ns, no significant difference. (c) Relative abundance at the genus level in different samples. Mono_Food, artificial feed alone without silkworms or feces for 2 days; Co_Food, artificial feed placed with silkworms and feces without antibiotics for 2 days. Three individuals per sample were used, to average one silkworm.

Thus, bacteria easily proliferate again during feeding (via silkworms and creeping) and storage. Therefore, artificial feed without antibiotics was consumed by 5th-instar silkworms for 2 days, and the remaining food was designated the Co_Food, which had traces of the feeding process (silkworm climbing and feces fermentation), simulating the actual production in the breeding factory. For comparison, artificial feed without antibiotics was placed alone under the same environmental conditions for 2 days and designated the Mono_Food group, with no feces or silkworm activity.

Compared to the Mono_Food group, the Co_Food group showed higher species richness (Fig. 3a) and lower community diversity (Fig. 3b; other indexes are shown in Table S1c). Specifically, the relative abundance of *Lactobacillus* and *Weissella* increased compared to those in the Mono_Food group, and that of *Gluconobacter* decreased (Fig. 3c). This suggests that silkworm climbing and feces enriched the bacteria in the feed and are conducive to the reproduction of lactic acid bacteria (LAB) [27] such as *Lactobacillus* and *Weissella*. The microbial distribution in the silkworm gut and different foods was further analyzed using a heat map (Fig. 4). Compared with the artificial feed, mulberry leaves introduced more microorganisms into the gut of the silkworms. Meanwhile, some microorganisms, such as *Lactobacillus*, *Gluconobacter*, and *Enterococcus*, occurred in great amounts in the artificial feed, whereas they were rarely or completely absent in the mulberry leaves.

The artificial feed with antibiotics, the Mono_Food(+) group, was placed alone for 1 month (due to the addition of antibiotics, it was not easily enriched) for 16S rRNA gene sequencing. The Venn diagram shows that the Mono_Food(+) group did not exhibit any unique genera (Fig. 5a). Among the eight genera shared by all food groups, more than 0.01% were LAB, such as *Lactobacillus*, *Weissella*, and *Enterococcus* (Fig. 5b). The structure of the Mono_Food(+) group at the OTU level closely resembled that of the Co_Food group and was significantly different from that of the Mono_Food group under natural conditions (Fig. 5c).

Artificial-feed-bred 5th-instar silkworm intestine (gut group, which was same as the Art and healthy groups mentioned above), fresh silkworm feces (feces group), and artificial feed (food group, which was the same as the Co_Food group mentioned above) after 2 days of cofeeding were used for 16S rRNA gene analysis. A Venn diagram analysis showed that 39 OTUs were shared by the three groups; that is, 39 OTUs ran through the food-gut-feces cycle (Fig. 6a). The food group had the fewest genera among the three groups and was mainly enriched with *Lactobacillus* and *Weissella*. The feces group was enriched not only with *Ralstonia*, *Rhodococcus*, and *Burkholderia-Caballeronia-Paraburkholderia*, which were also enriched in the Gut group, but also with *Weissella*, *Lactobacillus*, and *Bacillus* (Fig. 6b). Co-occurrence network analysis [28] showed that the relationship between the feces and gut groups was closer, and only *Lactobacillus* was related to all groups (Fig. 6c). The ternary-phase diagram showed the composition and distribution of dominant species in the different groups: *Lactobacillus* was mainly found in the food groups, and a small proportion occurred minimally in the other groups (Fig. 6d), while *Bacillus* appeared

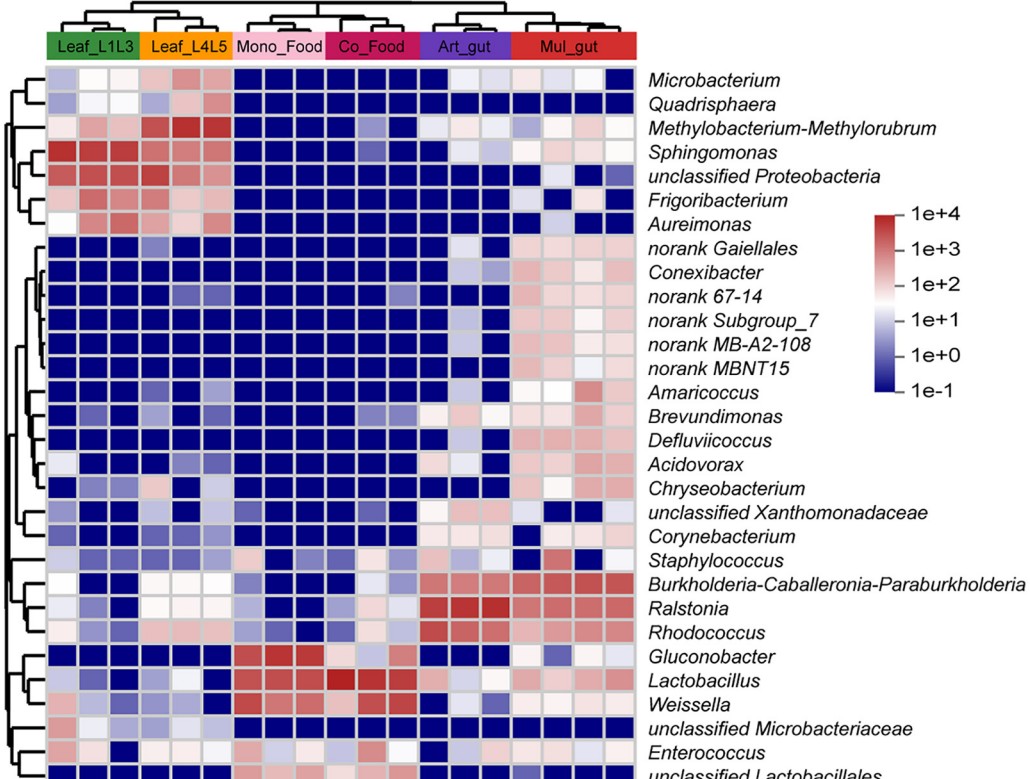

**FIG 4** Heat map of the microbiota in different foods and silkworm guts. Leaf_L1L3, young mulberry leaves suited for 1st- to 3rd-instar silkworms. Leaf_L4L5, old mulberry leaves suited for 4th- and 5th-instar silkworms. Art_gut, gut of the healthy silkworms fed artificial diet without antibiotics; Mul_gut, gut of the healthy silkworms fed mulberry leaves. Three individuals per sample were used, to average one silkworm.

mainly in the feces group. *Weissella* accounted for a minor difference between the food and feces groups, accounting for approximately 50%; *Ralstonia*, *Burkholderia-Caballeronia-Paraburkholderia*, and *Rhodococcus* were found in both the feces and gut groups, with a higher abundance in the gut group of approximately 60 to 70%. LEfSe showed that there were 13 significantly different genera between the groups (Fig. 6e): one unclassified genus belonging to *Lactobacillales* in the Food group and three genera in the feces group, namely, *Caulobacter*, *Lysinibacillus*, and *Brachybacterium*. Based on the function prediction analysis of PICRUSt2 (KEGG pathway level 3), the "Biosynthesis of amino acids," "ABC trans-

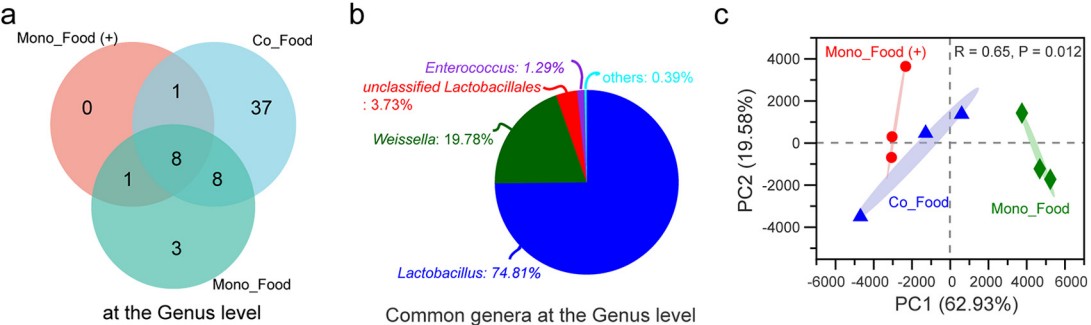

**FIG 5** Microbiota differences in food with antibiotics. (a) Venn diagram of the three groups at the OTU level. (b) Pie diagram of the common genera at the genus level. (c) PCA plot based on the structure of the microbial community at the OTU level. Mono_Food(+), artificial feed placed alone with antibiotics for 1 month (red circles); Co_Food, artificial feed placed with silkworms and feces without antibiotics for 2 days (blue triangles); Mono_Food, artificial feed alone for 2 days (green diamonds). The percentage of variation explained by each axis (PC1 and PC2) refers to the fraction of the total variance of the data explained by the constrained factor. Community variation is based on Bray-Curtis distances and an ANOSIM with 999 permutations. Three individuals per sample were used, to average one silkworm.

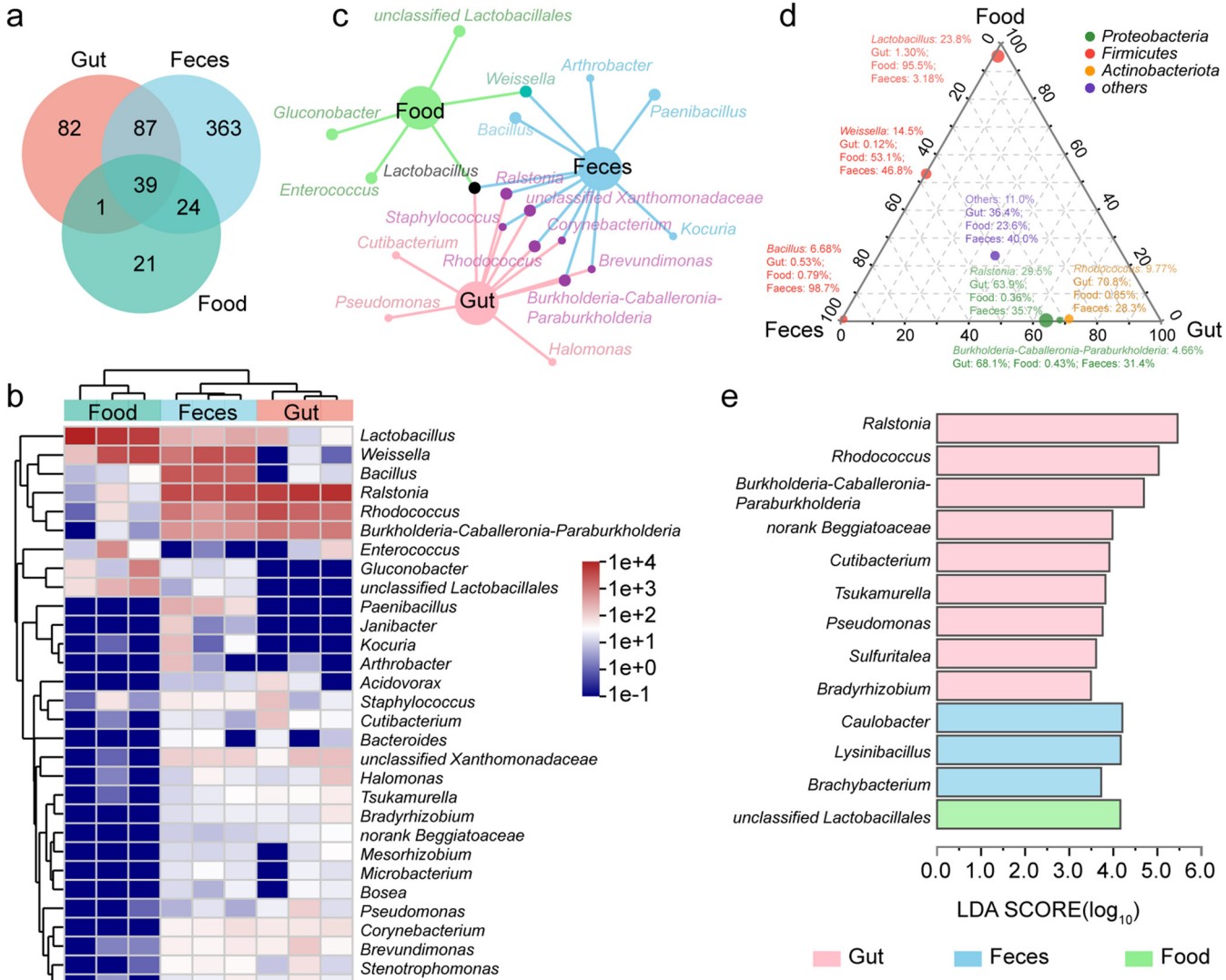

**FIG 6** Microbiota in food-gut-feces. (a) Venn diagram of three groups at the OTU level. (b) Heat map showing the relative abundance of the dominant taxa with respect to treatment. Hierarchical cluster analysis is based on Bray-Curtis distances using a complete-linkage method. Each sample was measured in triplicate. (c) Co-occurrence network analysis of the three groups. A node represents a species or group, and the edge represents the species included in the group. A larger species node indicates a higher degree of centrality of the node and increased importance of the node in the network. (d) Ternary-phase diagram. The three corners represent the three groups, the legend color represents the corresponding taxonomic phylum level, the circle represents the species under the phylum level, and the size of the circle represents the average relative abundance of the species. (e) Species difference analysis based on LEfSe at the genus level in different groups. Three individuals per sample were used, to average one silkworm.

porters," and "Carbon metabolism" pathways were less enriched in the food group than in the gut and feces groups (Fig. S2d).

As LAB may play an important role, the pH value of each component was measured. The pH of the intestinal juice decreased in the silkworms fed the artificial diet compared with that in silkworms fed mulberry leaves, and the pH decreased more significantly with the aggravation of disease (Fig. S7a). In the food and feces, the pH of the artificial feed was lower than that of mulberry leaves, and the pH of the cofeed was lower than that of the natural feed (Fig. S7b). Moreover, the pH of the feces from the silkworms fed the artificial diet was lower than that of the feces from silkworms fed mulberry leaves (Fig. S7c).

## DISCUSSION

**Intestinal disorders caused by food changes provide ideal conditions for opportunistic pathogens.** After silkworms were fed an artificial diet instead of mulberry leaves, the composition of the intestinal microbiome in the silkworms showed

considerable differences (Fig. 1): the number of species making up the microbiota decreased, and the structure was singular. From an ecological perspective, a decrease in biodiversity increases the likeliness of disruption to the community structure, which increases the possibility of disease transmission (29, 30). Our investigations showed that silkworms fed an artificial diet were more likely to acquire bacterial diseases than those fed a mulberry leaf diet. Generally, an opportunistic pathogen does not harm its host; however, when the host's resistance is low, it can cause disease (31). For example, *Staphylococcus aureus* can colonize the human nose asymptomatically and be transmitted from person to person without causing disease, thereby establishing long-term residence (32). There were no unique OTUs in the sick silkworms fed an artificial diet with antibiotics, and only a few strains were shared with the healthy silkworms (Fig. 2d). This uniform sick-phenotypic microbiota was dominated by *Enterococcus* (more than 95%) (Fig. 2c). Through microbiome analysis and inoculation experiments, we preliminarily determined the relationship between *Enterococcus* and bacterial diseases. *Enterococcus* meets the definition of colonizing opportunistic pathogenic bacteria. In the intestines of the healthy silkworms administered antibiotics, *Enterococcus* still composed a high proportion at more than 30%, and the other genera were mainly other LAB, significantly different from those of healthy silkworms not administered antibiotics (Fig. 2c). Antibiotic use during human infancy induces imbalances in the gut microbiota, commonly called dysbiosis (33), which can lead to various diseases. With the addition of antibiotics, the structure of the intestinal microbiota of the silkworms showed dramatic differences; the structure was single, and *Enterococcus* proliferated (Fig. 2). Even without antibiotics, silkworms fed the artificial diet showed the same trend as those fed the mulberry leaf diet (Fig. 1). The lack of key bacteria in the intestines significantly reduced diversity and led to dysbiosis, which created an environment for opportunistic *Enterococcus* strains to cause disease.

**Distribution of key dominant genera in each group.** Unlike mulberry leaves, the artificial feed had no source of primitive microbiota, as it was exposed to high temperatures. However, the first three dominant genera, *Ralstonia*, *Rhodococcus*, and *Burkholderia-Caballeronia-Paraburkholderia*, were the same between the two diet types (Fig. 1c), indicating that the common presence of these genera may be related to the host's own selection. Also, the fact that the dominant genera were the same in both groups may be responsible for the fact that there was no significant difference in the top 20 KEGG pathways (Fig. S2a). *Ralstonia* is often reported as the dominant genus in silkworms and has been mentioned as being related to plant diseases (34), which is consistent with the high contribution of the "Potentially pathogenic" pathway predicted by BugBase in this study (Fig. S6e). Therefore, it has been speculated that *Ralstonia* is obtained by the silkworms from the leaf surface of the host plant (35). However, in this study, silkworms fed the artificial diet were also enriched in *Ralstonia*, and the relative abundance was higher than that in silkworms fed mulberry leaves. Thus, plant foliage may not be the main source of *Ralstonia*, and there may be unknown factors that regulate its symbiosis with silkworms.

*Rhodococcus* biodegrades acetamide insecticides and is beneficial to detoxification in silkworms (36). Some *Rhodococcus* strains produce unique peptide loops that exhibit selective activity against *Mycobacterium smegmatis* and inhibit the growth of *Mycobacterium tuberculosis* (37). *Burkholderia-Caballeronia-Paraburkholderia* is a new taxonomic clade that accommodates many beneficial and environmental bacterial species, many of which are related to plants (promoting plant growth, symbiotic diazonium organisms, and free-living species), diazo nutrition, bioremediation, and antibiotic activity (38). These three dominant genera were significantly more abundant in the gut and feces of silkworms than in the food (artificial feed) (Fig. 6b), and because of this and the changes in the heat map produced by PICRUSt2, these genera were positively correlated with the "Biosynthesis of amino acids," "ABC transporters," and "Carbon metabolism" pathways (Fig. S2d). The existence of these genera may be beneficial to silkworms, as they assist in detoxification and bioremediation, and these results may provide a reference for silkworm probiotic products.

Those bacteria may be targeted and isolated for addition to diets to regulate the gut balance of *B. mori*.

The three above-mentioned genera occupied a dominant position in the gut of the silkworms and had a rich distribution in the feces group (Fig. 6b); however, they were reduced to no more than 1% in diseased silkworms and those fed the artificial feed with added antibiotics (whether healthy or not) (Fig. 2c). The dominant bacteria in the diseased silkworms in this study and in those fed the artificial feed with added antibiotics were *Enterococcus*, *Weissella*, *Lactobacillus*, and other LAB (39), which were widely found in feces and food samples (Fig. 6b). *Weissella* and *Lactobacillus* are LAB that can produce lactic acid and promote the fermentation of plant-derived food in the intestinal tract of animals to facilitate gastrointestinal absorption. These bacteria are often used as probiotics to ferment food and animal feed (27), which was consistent with the elevated representation of the "Starch and sucrose metabolism" pathway in the diseased silkworms (Fig. S2b). Additionally, some of these species act as opportunistic pathogens, and the presence of several antibiotic resistance-coding genes may increase the potential pathogenicity of some strains (40). In this study, *Weissella*, *Lactobacillus*, and other LAB were negatively correlated with health status and positively correlated with the presence of *Enterococcus* (Fig. 2g). Their excessive enrichment may change the intestinal environment and promote the proliferation of *Enterococcus*. The growth of these genera could be inhibited to regulate intestinal balance.

*Enterococcus* was closely associated only with "Mobile element containing" in the BugBase phenotype prediction (Fig. S6b), but it is an important human pathogen that causes various diseases, has strong drug resistance, and can survive under harsh conditions (41). In silkworms, it is often the dominant strain (42–44). Inoculation of silkworm larvae with *E. mundtii* may have a negative impact on the host response to chlorpyrifos (45). Additionally, *E. mundtii* has been identified as the pathogenic agent of flacherie in *B. mori* larvae reared on an artificial diet with chloramphenicol (46). There were four different OTUs of *Enterococcus* in this study; however, only OTU856 was highly abundant in all groups, particularly in the diseased silkworms (Table S2). The three strains of *E. mundtii* that were isolated all matched OTU856, could cause disease after inoculation, and had different pathogenic abilities (Fig. S5). There were also significant differences among the different strains of the same genus. As only a small region (394 bp) of the 16S rRNA gene was sequenced, species classification was limited, and the differential identification of strains was inadequate. Thus, in future, these data should be combined with metagenomic sequencing to contain more comprehensive information.

**Food-gut-feces microbial model influencing gut dysbiosis.** In this study, we found that there may be two factors that affect the changes in the intestinal microbiota of *B. mori*, which include differences in the feeding methods as well as the food itself (Fig. S8). An earlier study found that mulberry leaves can provide more sources of microorganisms, increasing the complexity and stability of microorganism intestinal structure and decreasing the likeliness of disruption (47). The feeding method of silkworms with an artificial diet was different from that with mulberry leaves. Artificial feed is a nutrient-rich medium and is directly exposed to the external environment. Based on the different properties of the host, different bacteria in the environment are attracted (48). The climbing and feeding behavior of silkworms and cofermentation of feces lead to enrichment of LAB such as *Lactobacillus* and *Weissella* in the artificial feed, leading to an emission of a sour odor and altering the living environment of the silkworms.

The change in abundance of the dominant bacteria is worthy of attention, as it is an indispensable regulator of intestinal homeostasis (49) and largely affects the function of the entire microbiota and the gut microenvironment, including acid accumulation and pH change (50). The silkworm gut is alkaline, facilitating the formation of suitable microbial groups (18). However, the artificial feed was more acidic than mulberry leaves (Fig. S7b), providing an acidic environment. Moreover, the differences in the pH values of the

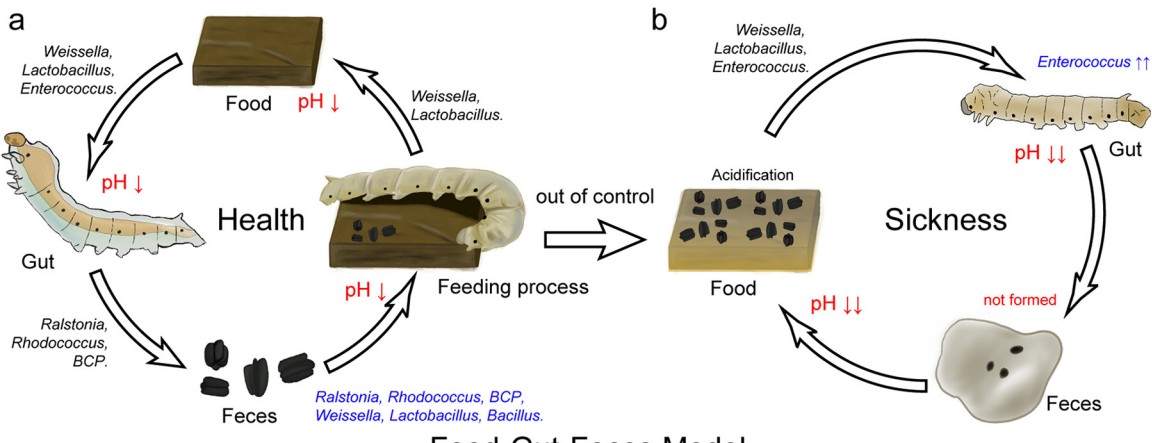

**FIG 7** Schematic diagram of the food-gut-feces microbial transfer model.

food, gut, and feces (Fig. S7) were consistent with the enrichment of LAB, with LAB-rich components exhibiting lower pH.

In summary, we propose a preliminary food-gut-feces microbial model (Fig. 7). First, the dominant genera in the gut of silkworms can also be enriched in the feces by excretion (Fig. 6b). Second, the host specificity of the artificial feed made it impossible to enrich these dominant genera but was more suitable for LAB such as *Lactobacillus* (Fig. 6b), as the pH of the food decreased (Fig. S7b). Third, these LAB can also be enriched in the feces (Fig. 6b), as the pH of the feces decreased (Fig. S7c). Fourth, the climbing of silkworms and co-fermentation of feces accelerated the enrichment of LAB in the artificial feed (Fig. 5 and Fig. S8), causing the pH to decrease (Fig. S7b). Fifth, after the ingestion of the artificial feed by silkworms, the pH of the intestinal tract dropped (Fig. S7a), which changed the intestinal environment and made it easier for *Enterococcus* and *Weissella* to colonize. Finally, when this change accumulated sufficiently, the gut ecology became unbalanced, and *Enterococcus* increased rapidly (Fig. 2c), which further disrupted the intestinal microbial homeostasis, damaged the host, and caused sickness in *B. mori* (Fig. S1 and S5).

Timely removal of feces, more frequent replacement of artificial feed, and maintenance of a clean environment could slow the process of gut microecological imbalance, and subsequent rearing did indeed improve, but illness was still more prevalent than with the mulberry leaf diet. The high microbial diversity and coevolution of the microbiota in mulberry-fed silkworms make them more adaptable to the nutrients of mulberry leaves. Sudden changes to dietary habits or the simple microbial structure of silkworms, such as by switching to an artificial diet, can make it difficult to maintain a balance. To address this issue in the future, key target bacteria could be added, and excess LAB could be inhibited to adjust the balance and could also be indirectly regulated by controlling the intestinal environment. The differential bacterial genera screened by LDA may also provide clues. During the domestication of livestock, such as pigs, cattle, and sheep, their food has undergone industrialization and simplification, and the health of the hosts is poorer than that of free-range animals (51). Humans are also prone to diseases such as gastrointestinal discomfort after changes in their long-term habitual diet. Owing to the difficulty in obtaining materials, a long acquisition period, and difficulties with research, our present study may provide valuable information for the successful treatment of gastrointestinal discomfort in humans.

**Conclusion.** After habitual dietary changes, the body often appears unwell. Using the domestic silkworm *B. mori* as a model, we changed its traditional mulberry leaf dietary habit to an artificial feed and found that the silkworms showed more maladaptive phenomena under the artificial diet, and particularly increased incidence of disease. From the perspective of the microbiome, a systematic study was conducted on *B. mori*

larvae fed artificial diets; the differences in the intestinal microbiome were described in detail between the artificial and mulberry leaf diets, and the reasons were explored. The following results were obtained. (i) Mulberry leaves could provide a richer microbiota and help maintain the stability of the gut, whereas the artificial feed made the intestinal microenvironment more acidic due to the accumulation of LAB, affecting the intestinal microbes, which further affected the physicochemical properties of the intestinal juice and led to nutrition malabsorption and a reduced immune-disease resistance. (ii) In this process, feces provided abundant lactic acid-secreting bacteria for the artificial feed, which accelerated its acidification process. (iii) The agents of bacterial diseases that often appeared in the artificial diet were initially targeted, which may provide guidance for future disease prevention and control. (iv) The dominant genera and their distributions were analyzed in detail, and some potential key genera that could regulate the balance of intestinal microbiota were screened out, which provided insights for improving the artificial feed for breeding silkworm probiotic products. (v) Finally, a food-gut-feces microbial model was established to provide a reference for future studies on different species with similar difficulties.

## MATERIALS AND METHODS

**Preparation of the artificial diet.** The artificial diet used in this study was independently developed by our team (Biological Science Research Center, Southwest University, China). Artificial diet powder was prepared using 30% mulberry leaf powder (from a market in Chongqing, China), 35% defatted soybean meal (from a market in Chongqing, China), 20% corn flour (from a market in Chongqing, China), 5% vitamins and minerals (Sangon Biotech, Shanghai, China), and 10% others. Additionally, 0.01% florfenicol (Sangon Biotech, Shanghai, China) was added to the artificial diet for the antibiotic group. The diet powder was mixed thoroughly with water at a 1:3 ratio of powder to water. This mixture was then steamed at 100°C for 30 min, mixed, and stored at 4°C until use.

**Silkworm rearing.** The *B. mori* strain Liangguang II was maintained at Southwest University, Beibei, China. Fresh mulberry leaves were collected from the Legen Robust Modern Sericulture Research Base (Tongliang, Chongqing, China). The egg surface was disinfected per the following protocol: 3% formaldehyde for 5 min, sterile washing three times, 75% ethanol for 2 min, and finally sterile washing an additional three times. The silkworms were fed under standard conditions (25 $\pm$ 1°C, 70% $\pm$ 5% humidity with a photoperiod of 12 h/12 h light/dark) in a greenhouse with a fresh air system. Indicators, including phenotypic data (weight and instar duration), pH value, disease incidence, and microbial diversity were assessed. All experiments were conducted in triplicate with at least 300 individuals per treatment; there were three individuals per sample (100 individuals per sample for weight and pH value) to average one silkworm to reduce individual differences. The experimental design and grouping are detailed in the supplemental materials and methods and Fig. S9.

**Sample collection and sequencing.** Whole-gut tissue was dissected from 5th-instar silkworms after the surface had been disinfected with 70% ethanol and was directly transferred into a 2-mL sterile cryogenic vial that was placed on ice. Other samples, such as the artificial feed, fresh mulberry leaves, and feces, were also directly transferred. Total nucleic acid was extracted using an E.Z.N.A. soil DNA kit (Omega Bio-Tek, Norcross, GA, USA). To avoid chloroplast interference from mulberry leaves and in the gut of silkworms fed on mulberry leaves, we utilized primers 799F and 1193R to amplify a region of the 16S rRNA gene (52, 53).

Two rounds of amplification were completed in an ABI GeneAmp 9700 PCR thermocycler (Applied Biosystems, Waltham, MA, USA) using forward primer 799F (5'-AACMGGATTAGATACCCKG-3') and reverse primer 1392R (5'-ACGGGCGGTGTGTRC-3') in the first round, with an annealing temperature of 55°C for 27 cycles to avoid the host information and improve the specificity, and forward primer 799F and reverse primer 1193R (5'-ACGTCATCCCCACCTTCC-3'), with an annealing temperature of 55°C for 13 cycles to amplify again based on the first round. The target fragment in the second round was short enough to be detected by the sequencing platform (Illumina MiSeq PE300 platform; Illumina, San Diego, CA, USA) (54). The PCR mixtures contained 4 $\mu$L of 5$\times$ TransStart FastPfu buffer (TransGen, Beijing, China), 2 $\mu$L of 2.5 mM deoxyribonucleotide triphosphate, 0.8 $\mu$L of forward primer (5 $\mu$M), 0.8 $\mu$L of reverse primer (5 $\mu$M), 0.4 $\mu$L of TransStart FastPfu DNA polymerase (TransGen), 10 ng of template DNA, and double-distilled water up to a volume of 20 $\mu$L. PCRs were performed in triplicate. The PCR products were extracted from 2% agarose gels (Biowest, Nuaillé, France), purified using an AxyPrep DNA gel extraction kit (Axygen Biosciences, Union City, CA, USA) according to the manufacturer's instructions, and quantified using a Quantus fluorometer (Promega, Madison, WI, USA). Purified amplicons were pooled in equimolar concentrations and paired-end sequenced on an Illumina MiSeq PE300 platform (Illumina), according to the standard protocols from Majorbio Bio-Pharm Technology Co. Ltd. (Shanghai, China).

**Sequencing data processing.** The raw 16S rRNA gene sequencing reads were demultiplexed, quality filtered using fastp version 0.20.0 (55), and merged using FLASH version 1.2.7 (56), according to a standard protocol (57).

The data were analyzed using the Majorbio Cloud platform (www.majorbio.com). OTUs were clustered at 97% similarity using UPARSE version 7.1 (58). The taxonomy of each OTU representative

sequence was analyzed using RDP Classifier version 2.2 (59) against the 16S rRNA database (Silva v138) using a confidence threshold of 0.7 (Table S3). Rarefaction curves (60) were prepared using enough sequencing data to produce a flat curve (Fig. S10), meaning that the amount of sequencing data was large enough to reflect the vast majority of microbial diversity information in the sample.

**Isolation and identification of gut bacteria.** Three *Enterococcus mundtii* strains (CQJ-1, CQJ-2, and CQJ-3) and *Staphylococcus sciuri* SH were isolated and purified from the guts of sick silkworms fed the artificial diet without antibiotics. Bacterial suspensions were obtained by culturing cells in 300 mL of lysogeny broth medium (10 g tryptone, 5 g yeast extract, and 10 g NaCl L$^{-1}$) on a rotary shaker (150 $\times$ *g*) at 37°C for 12 h. The bacterial cells were collected by centrifugation (4,500 $\times$ *g* for 5 min at 4°C) three times with sterile H$_2$O to compare the phenotypes of the bacterial cell. To preliminarily confirm the strain of bacteria isolated from the sample, PCRs (94°C for 10 min followed by 34 cycles at 94°C for 30 s, 56°C for 30 s, and 72°C for 90 s, with a final extension at 72°C for 10 min) were performed with the universal 16S rRNA gene primers 27F (5′-AGAGTTTGATCCTGGCTCAG-3′) and 1492R (5′-TACGGTTACCTTGTTACGACTT-3′). The PCR products were sequenced by BGI (Shenzhen, China), and the resultant sequences were compared with those in the GenBank database using BLAST. A molecular evolutionary tree for the three *E. mundtii* strains and the four OTUs of *Enterococcus* was built using the same region of microbial diversity (799F and 1193R) via the neighbor joining method with a bootstrap value of 1,000 in Molecular Evolutionary Genetics Analysis version 11 (61).

**Evaluation of silkworms after inoculation with gut bacteria.** Fourth-instar silkworms fed the artificial diet were inoculated with 20-$\mu$L bacterial suspensions that were resuspended and diluted in deionized water (1 $\times$ 10$^9$ CFU·mL$^{-1}$). The same volume of sterile water was used as a control group. We then compared the phenotype of 5th-instar diseased silkworms inoculated with different strains. Next, we inoculated 50 $\mu$L of intestinal fluid diluent (serial dilution 10$^3$ times) from diseased silkworms after inoculation on *Enterococcus* chromogenic medium (HB7014-1; Hopebiol, Shandong, China) at 37°C for 48 h in the dark to compare the growth of *Enterococcus* in guts inoculated with different strains.

**Statistical analysis.** In the figures, the weights and pHs of the silkworms are presented as means and standard errors (SE). Values were compared using one-way analysis of variance (ANOVA) followed by Bonferroni's *post hoc* test using the Statistical Package for the Social Sciences, v22.0 (SPSS, Chicago, IL, USA). Means among treatments were considered significantly different when the probability (*P* value) was less than 0.05.

For the statistical analyses of 16S rRNA gene amplicon data, alpha diversity, including the Sobs, Shannon, Ace, Chao1, Simpson, coverage, and other indexes, was calculated using mothur (version v.1.30.2; https://mothur.org/wiki/calculators/). Significant differences were assessed by Student's *t* test. Community composition analysis (bar and pie diagrams), Venn diagram and heat map construction, and PCA were carried out using R version 3.3.1. A nonparametric Kruskal-Wallis test was used when the data were not normally distributed. The Bray-Curtis dissimilarity metric and analysis of similarities (ANOSIM) with 999 permutations were performed when comparing groups. Those analyses were performed using the Majorbio Cloud Platform (www.majorbio.com) (62).

Species difference analysis based on LEfSe was conducted at the genus level using the LEfSe tool of the Galaxy application (http://huttenhower.sph.harvard.edu/galaxy/root?tool_id=lefse_upload). We considered taxa with LDA scores of >2 and *P* values of <0.05 to be significantly different. Co-occurrence network analysis reflected the coexistence relationship of genera in environmental groups. The species-related network (one-way network analysis) reflected interactions between genera based on genus-to-genus correlations. The two network analyses were performed in Python version 2.7 using the NetworkX package. A ternary-phase diagram was created using GGTERN (http://www.ggtern.com/) to show the composition and distribution of dominant genera in the different groups. Microbiome functions were predicted using PICRUSt2 and BugBase from the 16S rRNA data, and a KO functional abundance statistics table (Table S4) was prepared.

**Ethics approval.** This study only used silkworms as biological experimental materials. Being invertebrates, silkworms are not protected by any legislations from experimental use. All silkworm pupae removed from cocoons were disposed of as biological waste.

**Data availability.** 16S rRNA gene amplicon raw sequencing data were deposited in the National Center for Biotechnology Information Short Read Archive, BioProject PRJNA791511, under the accession numbers SRR17307164 to SRR17307203.

## SUPPLEMENTAL MATERIAL

Supplemental material is available online only.
**SUPPLEMENTAL FILE 1**, PDF file, 2.8 MB.
**SUPPLEMENTAL FILE 2**, XLS file, 0.04 MB.
**SUPPLEMENTAL FILE 3**, XLS file, 0.03 MB.
**SUPPLEMENTAL FILE 4**, XLS file, 0.6 MB.
**SUPPLEMENTAL FILE 5**, XLS file, 2.1 MB.

## ACKNOWLEDGMENTS

This study was supported by the National Natural Science Foundation of China (grant no. 32030103) and Municipal Graduate Student Research Innovation Project of Chongqing (grant no. CYB21134).

We declare that the research was conducted in the absence of any commercial or financial relationships that could be construed as a potential conflict of interest.

L.Q. designed and performed the experiment. Q.X. conducted the experiment. G.S. provided the artificial feed and silkworm eggs. G.S., J.Q., D.Q., and J.W. helped manufacture the artificial feed. L.Q. and J.Q. performed the bacterial isolation and inoculation experiments. L.Q., J.Q., and Y.S. analyzed the data, wrote the original draft of the manuscript, and performed the lab work. L.Q., D.Q., J.W., and Y.S. performed silkworm rearing and dissection. L.Q., Y.C., and Q.X. reviewed and edited the manuscript. Q.X., Y.C., P.Z., and L.Q. obtained the funding. All authors read and approved the final manuscript.

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
