## [Reviewer comments · Microbiology Spectrum]

Microbiology Spectrum

Effects of Food–Gut–Feces Microbial Circulation Transfer on the Health of *Bombyx mori*

Lijun Qin, Junpeng Qi, Guanwang Shen, Daoyuan Qin, Jinxin Wu, Yuwei Song, Yang Cao, Ping Zhao, and Qingyou Xia

Corresponding Author(s): Qingyou Xia, Biological Science Research Center of Southwest University; State Key Laboratory of Silkworm Genome Biology

Review Timeline:

Submission Date:	June 23, 2022
Editorial Decision:	July 26, 2022
Revision Received:	September 23, 2022
Editorial Decision:	October 1, 2022
Revision Received:	October 6, 2022
Accepted:	October 7, 2022

Editor: Chengshu Wang

Reviewer(s): Disclosure of reviewer identity is with reference to reviewer comments included in decision letter(s). The following individuals involved in review of your submission have agreed to reveal their identity: Guanhong Wang (Reviewer #2)

Transaction Report:

DOI: <https://doi.org/10.1128/spectrum.02357-22>

July 26, 2022

Prof. Qingyou Xia
Biological Science Research Center of Southwest University; State Key Laboratory of Silkworm Genome Biology
State Key Laboratory of Silkworm Genome Biology, Southwest University
Chongqing 400715
China

Re: Spectrum02357-22 (Effects of Food-Gut-Feces Microbial Circulation Transfer on the Health of *Bombyx mori*)

Dear Prof. Qingyou Xia:

Thank you for submitting your manuscript to Microbiology Spectrum. Your manuscript has been reviewed by two experts in this field. As you will see, they both agree that it is an interesting study but requires additional works. When submitting the revised version of your paper, please provide (1) point-by-point responses to the issues raised by the reviewers as file type "Response to Reviewers," not in your cover letter, and (2) a PDF file that indicates the changes from the original submission (by highlighting or underlining the changes) as file type "Marked Up Manuscript - For Review Only". Please use this link to submit your revised manuscript - we strongly recommend that you submit your paper within the next 60 days or reach out to me. Detailed instructions on submitting your revised paper are below.

Link Not Available

Sincerely,

Chengshu Wang

Journals Department
Reviewer comments:

Reviewer #1 (Comments for the Author):

The authors present a study to explore the relationship between food change and the balance of intestinal microbes of the domestic silkworm. This study is exquisitely designed and displays some interesting results and make some inferences according to these results. While, the data amount is not so much and some other problems are still existing about this manuscript.

Major problems:

1. Many grammatical mistakes and awkward phrasings can be found in the manuscript. For example, Line 92-93: "which is suitable for extracting DNA of plants, feed, and feces for soil with numerous impurities and high heterogeneity". So, I advise the manuscript need to be thoroughly checked and modified by a native English speaker.

2. Line 56, the references 12 and 13 are reviews of gut microbiota in vertebrate, the manuscript cite these two references here to support your viewpoint of silkworms is very unsuitable.
3. Line67-68, "Our study findings will provide insights for similar research in other animals, even humans, and technical references". I do not know if the workload and technology used in this work can support this statement.
4. Line 153, "A heatmap was created using the vegan package in R version 3.3.1". Vegan is a package for statistical analysis but not for plot, authors need to check if this sentence is suitable. Similar, only R can not process the PCA plot, so authors need to write which package are used to create these PCA plot. All in all, the method section of this manuscript is very cumbersome.
5. Line169-170. "Analyses were performed using the Statistical Package for the Social Sciences, v22.0 (SPSS, Chicago, IL, USA) or the free online Majorbio Cloud Platform (www.majorbio.com)". I do not know what this sentence mean, which analyses were performed using SPSS and which were used Majorbio Cloud Platform?
6. Line 181, the Shannon is an index for evenness, so only use this index stand for community diversity is debatable.
7. Line310-326, this paragraph supposed some functions of the bacteria. But in my opinion, even if the metagenome is hard to access, the function prediction works performed by PICRUSt2 or other software are still feasible.
8. According to the ICPN (International Code of Nomenclature of Prokaryotes, <https://doi.org/10.1099/ijsem.0.000778>), all the categories of Prokaryotes need to be italic including family and other higher categories.

minor problems:

Line 23: " enriched with" delete with.

Line 39: " biota " or microbiota?

Line 40: "a" to " the".

Line 41: " Bifidobacterium longum and Bacteroidetes ", I do not think parallelize a species and a phylum at the same time is a good writing.

Line 45: " in" to " of".

Line 50: " a simple diet " to " simple diets ".

Line 61: " adapt to " to " adapt ".

Line 130: " 16S ", this is very nonstandard, authors should modify to 16S rRNA gene.

Line 147: " 16S rRNA statistical analyses ", the 16S rRNA gene amplicon data is analyzed, but not the RNA.

Line 252: " many microorganisms in the mulberry leaves occurred in small amounts or not at all with the artificial feed " this sentence is very awkward, please check and modify.

Reviewer #2 (Comments for the Author):

Major:

1. it seems the samples for some treatment are only 3? it is better to increase the sample size like in figure 3 C, the variation within treatments are big.

2. The author found some core microbiota or specific genus bacteria related to the phenotype, right now it is association and it may be better to isolate some of the specific bacteria and test their potential function.

Minor:

Fig. S2 re-scale the x-axis, the current one is hard to see how many reads numbers are enough for the following analysis.

Fig. S4, are there any assays that can quantify the current phenotype, or add some bodyweight phenotype that is easy to quantify?

Fig1, a Y-axis community diversity needs to be more specific, e x-axis between changes to "Art to Mul", Art changes to "Art to Art"..., recommend doing statistics between "between and Art", "between and Mul".

Fig3, a and b please label whether there are significant or not.

Staff Comments:

Preparing Revision Guidelines

Please return the manuscript within 60 days; if you cannot complete the modification within this time period, please contact me. If you do not wish to modify the manuscript and prefer to submit it to another journal, please notify me of your decision immediately so that the manuscript may be formally withdrawn from consideration by Microbiology Spectrum.

Comments on the manuscript Spectrum02357-22

The authors present a study to explore the relationship between food change and the balance of intestinal microbes of the domestic silkworm. This study is exquisitely designed and displays some interesting results and make some inferences according to these results. While, the data amount is not so much and some other problems are still existing about this manuscript.

Major problems:

1. Many grammatical mistakes and awkward phrasings can be found in the manuscript. For example, Line 92-93: “which is suitable for extracting DNA of plants, feed, and feces for soil with numerous impurities and high heterogeneity”. So, I advise the manuscript need to be thoroughly checked and modified by a native English speaker.
2. Line 56, the references 12 and 13 are reviews of gut microbiota in vertebrate, the manuscript cite these two references here to support your viewpoint of silkworms is very unsuitable.
3. Line67-68, “Our study findings will provide insights for similar research in other animals, even humans, and technical references”. I do not know if the workload and technology used in this work can support this statement.
4. Line 153, “A heatmap was created using the vegan package in R version 3.3.1”. Vegan is a package for statistical analysis but not for plot, authors need to check if this sentence is suitable. Similar, only R can not process the PCA plot, so authors need to write which package are used to create these PCA plot. All in all, the method section of this manuscript is very cumbersome.
5. Line169-170. “Analyses were performed using the Statistical Package for the Social Sciences, v22.0 (SPSS, Chicago, IL, USA) or the free online Majorbio Cloud Platform

(www.majorbio.com)". I do not know what this sentence means, which analyses were performed using SPSS and which were used Majorbio Cloud Platform?

6. Line 181, the Shannon is an index for evenness, so only use this index stand for community diversity is debatable.
7. Line 310-326, this paragraph supposed some functions of the bacteria. But in my opinion, even if the metagenome is hard to access, the function prediction works performed by PICRUSt2 or other software are still feasible.
8. According to the ICPN (International Code of Nomenclature of Prokaryotes, <https://doi.org/10.1099/ijsem.0.000778>), all the categories of Prokaryotes need to be italic including family and other higher categories.

minor problems:

Line 23: " enriched with" delete with.

Line 39: " biota " or microbiota?

Line 40: "a" to " the".

Line 41: " *Bifidobacterium longum* and Bacteroidetes ", I do not think parallelize a species and a phylum at the same time is a good writing.

Line 45: " in" to " of".

Line 50: " a simple diet " to " simple diets ".

Line 61: " adapt to " to " adapt ".

Line 130: " 16S ", this is very nonstandard, authors should modify to 16S rRNA gene.

Line 147: " 16S rRNA statistical analyses ", the 16S rRNA gene amplicon data is analyzed, but not the RNA.

Line 252: " many microorganisms in the mulberry leaves occurred in small amounts or not at all with the artificial feed " this sentence is very awkward, please check and modify.

Responses to Reviewers' Comments

For Reviewer #1

Comments:

The authors present a study to explore the relationship between food change and the balance of intestinal microbes of the domestic silkworm. This study is exquisitely designed and displays some interesting results and make some inferences according to these results. While, the data amount is not so much and some other problems are still existing about this manuscript.

Response: Thank you very much for your favorable consideration.

Major problems:

1. Many grammatical mistakes and awkward phrasings can be found in the manuscript. For example, Line 92-93: "which is suitable for extracting DNA of plants, feed, and feces for soil with numerous impurities and high heterogeneity". So, I advise the manuscript need to be thoroughly checked and modified by a native English speaker.

Response: Many thanks for your suggestion. Our manuscript has been polished based on the language editing results from the Cactus Communications Private Limited (<https://www.editage.cn/>).

2. Line 56, the references 12 and 13 are reviews of gut microbiota in vertebrate, the manuscript cite these two references here to support your viewpoint of silkworms is very unsuitable.

Response: As you suggested, we have replaced two suitable references.

The references [1, 2] cited here are as follows:

[1] Groussin M, Mazel F, Alm E J (2020) Co-evolution and co-speciation of host-gut bacteria systems. *Cell Host & Microbe* 28: 12-22. <https://doi.org/10.1016/j.chom.2020.06.013>

[2] Dong HL, Zhang SX, Chen ZH, Tao H, Li X, Qiu JF, Cui WZ, Sima YH, Cui WZ, Xu SQ (2018) Differences in gut microbiota between silkworms (*Bombyx mori*) reared on fresh mulberry (*Morus alba* var. *multicaulis*) leaves or an artificial diet. *RSC Adv* 8: 26188-26200. <https://doi.org/10.1039/c8ra04627a>

3. Line 67-68, "Our study findings will provide insights for similar research in other animals, even humans, and technical references". I do not know if the workload and

technology used in this work can support this statement.

Response: Thank you very much for your suggestion. We have revised.

4. Line 153, "A heatmap was created using the vegan package in R version 3.3.1". Vegan is a package for statistical analysis but not for plot, authors need to check if this sentence is suitable. Similar, only R can not process the PCA plot, so authors need to write which package are used to create these PCA plot. All in all, the method section of this manuscript is very cumbersome.

Response: Many thanks for your suggestion. We used the vegan package in R version 3.3.1 for statistical analysis, and the heatmap and PCA graphs were done on the Majorbio Cloud Platform (www.majorbio.com) [1]. According to your comments, we have modified the Methods section.

The references cited here are as follows:

[1] Ren Y, Yu G, Shi CP, Liu LM, Guo Q, Han C, Zhang D, Zhang L, Liu BX, Gao H, Zeng J, Zhou Y, Qiu YH, Wei J, Luo YC, Zhu FJ, Li XJ, Wu Q, Li B, Fu WY, Tong YL, Meng J, Fang YH, Dong J, Feng YT, Xie SC, Yang QQ, Yang H, Wang Y, Zhang JB, Gu HD, Xuan HD, Zou GQ, Luo C, Huang L, Yang B, Dong YC, Zhao JH, Han JC, Zhang XL, Huang HS (2022) Majorbio Cloud: A One-Stop, Comprehensive Bioinformatic Platform for Multiomics Analyses. *iMeta* e12. <https://doi.org/10.1002/imt2.12>

5. Line 169-170. "Analyses were performed using the Statistical Package for the Social Sciences, v22.0 (SPSS, Chicago, IL, USA) or the free online Majorbio Cloud Platform (www.majorbio.com)". I do not know what this sentence means, which analyses were performed using SPSS and which were used Majorbio Cloud Platform?

Response: Many thanks for your suggestion. We have revised the **M&M section 2.7** for simplicity and clarity.

6. Line 181, the Shannon is an index for evenness, so only use this index stand for community diversity is debatable.

Response: According to your comments, we have added other indices reflecting community richness (sobs, jack, bootstrap), evenness (simpson, shannon, heip, smithwilson), diversity (simpson, npshannon, bergerparker, invsimpson, qstat) to better assess microbial Alpha diversity. The new results have been included into the revised **Supplementary Table S3**, relative revised **Fig. 1, 2, and 3**, and modified the related description in the revised manuscript.

These indexes tended to be similar and did not affect the results, meanwhile, the Shannon index was more common and there were many references [1-3] about gut

microbiota use the Shannon index to assess community diversity. For the simplicity and clarity, the figures were still formed by Shannon and Chao1 indexes, and other indexes can be viewed in the revised **Supplementary Table S3**

The references cited here are as follows:

- [1] Chen B, Du K, Sun C, Vimalanathan A, Liang X, Li Y, Wang B, Lu X, Li L, Shao Y (2018) Gut bacterial and fungal communities of the domesticated silkworm (*Bombyx mori*) and wild mulberry-feeding relatives. ISME J 12: 2252-2262. <https://doi.org/10.1038/s41396-018-0174-1>
- [2] Wang H, Zhang JY, Wang XM, Hu HL, Xia RX, Li Q, Zhu XW, Wang TM, Liu YQ, Qin L (2020) Comparison of bacterial communities between midgut and midgut contents in two silkworms, *Antheraea pernyi* and *Bombyx mori*. Sci Rep 10: 12966. <https://doi.org/10.1038/s41598-020-69906-y>
- [3] Gandotra S, Kumar A, Naga K, Bhuyan PM, Gogoi DK, Sharma K, Subramanian S (2018) Bacterial community structure and diversity in the gut of the muga silkworm, *Antheraea assamensis* (Lepidoptera: Saturniidae), from India. Insect molecular biology 27: 603-619. <https://doi.org/10.1111/imb.12495>

7. Line310-326, this paragraph supposed some functions of the bacteria. But in my opinion, even if the metagenome is hard to access, the function prediction works performed by PICRUST2 or other software are still feasible.

Response: Thank you very much for your suggestion. We had tried metagenome, but it was difficult to obtain. As you suggested, we have supplemented the functional prediction results by PICRUST2 and BugBase. Accordingly, we have included these new results into the revised **supplementary Fig. S4, S5, and S8** and modified the related description in the revised manuscript.

8. According to the ICPN (International Code of Nomenclature of Prokaryotes, <https://doi.org/10.1099/ijsem.0.000778>), all the categories of Prokaryotes need to be italic including family and other higher categories.

Response: Many thanks for pointing this out. We have revised.

Minor problems:

Line 23: " enriched with" delete with.

Line 39: " biota " or microbiota?

Response: Thank you for pointing out this. We have fixed the error.

Line 40: "a" to " the".

Response: Many thanks for your suggestion. Grammar Editing Company represents: If this diet had been mentioned previously in the Introduction, then you could write “the”, but here, “a” is appropriate.”

Line 41: " Bifidobacterium longum and Bacteroidetes ", I do not think parallelize a species and a phylum at the same time is a good writing.

Response: Many thanks for your suggestion. We have changed the reference [1] and revised the sentence as “feeding a high-fat diet caused a significant reduction in diversity and overall structural shifts in bacterial communities in adult mice, such as a decrease in *Bacillus bifidus*”

The references cited here are as follows:

[1] Zhang CH, Zhang MH, Pang XY, Zhao YF, Wang LH, Zhao LP (2012) Structural resilience of the gut microbiota in adult mice under high-fat dietary perturbations. ISME J 6: 1848-1857. <https://doi.org/10.1038/ismej.2012.27>

Line 45: " in" to " of".

Line 50: " a simple diet " to " simple diets ".

Response: Thank you for pointing out this. We have fixed the error.

Line 61: " adapt to " to " adapt ".

Response: Many thanks for your suggestion. Grammar Editing Company represents: This word is necessary here. You cannot write “to adapt changes” because this would mean the silkworms were changing the changes rather than changing themselves in response to changes, which is expressed by “adapt to”.

Line 130: " 16S ", this is very nonstandard, authors should modify to 16S rRNA gene.

Response: Thank you for pointing out this. We have fixed the error.

Line 147: " 16S rRNA statistical analyses ", the 16S rRNA gene amplicon data is analyzed, but not the RNA.

Response: We thank the reviewer for pointing this out. We have revised.

Line 252: " many microorganisms in the mulberry leaves occurred in small amounts or not at all with the artificial feed " this sentence is very awkward, please check and modify.

Response: Many thanks for pointing this out. We have revised.

For Reviewer #2

Comments:

Major problems:

1. it seems the samples for some treatment are only 3? it is better to increase the sample size like in figure 3 C, the variation within treatments are big.

Response: Thank for your suggestion. All experiments and materials were repeated for at least 3 batches, and each batch had at least 300 silkworms for each treatment (5th instar). At the same time, each sample was averaged multiple individuals (100 individuals per sample for weight and pH value and three individuals per sample for others) to reduce individual differences. The data presented in this manuscript was from a batch that was relatively stable in all respects. For clarity, we supplemented this into **M&M section 2.2** and **Figure Legends**.

Moreover, the variation in the microbiome are relatively large, meanwhile, the individual differences of the silkworms fed on artificially diets are also greatly, which growth and development are uneven. This is another difficulty we are overcoming.

2. The author found some core microbiota or specific genus bacteria related to the phenotype, right now it is association and it may be better to isolate some of the specific bacteria and test their potential function.

Response: Many thanks for your suggestion. We are doing this work, it is just not systematic and waiting for more results.

Minor problems:

Fig. S2 re-scale the x-axis, the current one is hard to see how many reads numbers are enough for the following analysis.

Response: Many thanks for pointing this out. We have revised. We have added an enlarged view of the X-axis on the left.

Fig. S4, are there any assays that can quantify the current phenotype, or add some bodyweight phenotype that is easy to quantify?

Response: We agree with the reviewer that further elaborating on this point using new assays would be helpful. However, although we often observe this disease, the specific degree and weight of the disease are affected by many factors, such as silkworms (artificial diets) individual development is not neat, manual operation after the initial disease, etc. We can only briefly describe a dynamic process of its disease: "At first the body turned slightly yellow, then the tail decayed, and the body shortened.

At the time of death, the abdomen swelled slightly and the whole body was limp.”
And very sorry that we did not describe it in the **Fig. S4 legend (now changed to Fig S6)** (in the main text we did), we have now supplemented it.

Fig1, a Y-axis community diversity needs to be more specific, e x-axis between changes to "Art to Mul", Art changes to "Art to Art"., recommend doing statistics between "between and Art", "between and Mul".

Response: Many thanks for your suggestion. We have revised.

“Between” (now “Art to Mul”) represents the distance value of the difference between the groups, and the remaining boxes represent the distance value of the difference within the group. This analysis is to test the grouping difference, and there is only one *P* value (other values are the same), which represents whether the difference between groups is greater than the difference within the group (whether the grouping mode we have established is appropriate). The result was $P = 0.021$, the grouping was suitable, we have added to the revised **Fig. 1e**.

Fig3, a and b please label whether there are significant or not.

Response: We thank the reviewer for pointing this out. We have revised.

October 1, 2022

Prof. Qingyou Xia
Biological Science Research Center of Southwest University; State Key Laboratory of Silkworm Genome Biology
State Key Laboratory of Silkworm Genome Biology, Southwest University
Chongqing 400715
China

Re: Spectrum02357-22R1 (Effects of Food-Gut-Feces Microbial Circulation Transfer on the Health of *Bombyx mori*)

Dear Prof. Qingyou Xia:

Link Not Available

Sincerely,

Chengshu Wang

Journals Department
Reviewer comments:

Reviewer #1 (Comments for the Author):

Most questions I concerned before have been solved in the current manuscript. But I still have one question, in the sentence 114-115, we can find that "There was no significant difference between the Art and Mul groups in the KEGG pathway level 3 heatmap". But from figure one, we can find that the composition of microbial communities in Art and Mul groups are significantly different. It is an interesting phenomenon, so I think it is better to have some discussion words in the discussion part.

Staff Comments:

Preparing Revision Guidelines

Please return the manuscript within 60 days; if you cannot complete the modification within this time period, please contact me. If you do not wish to modify the manuscript and prefer to submit it to another journal, please notify me of your decision immediately so that the manuscript may be formally withdrawn from consideration by Microbiology Spectrum.

Comments on the manuscript Spectrum02357-22R1

Most questions I concerned before have been solved in the current manuscript. But I still have one question, in the sentence 114-115, we can find that “There was no significant difference between the Art and Mul groups in the KEGG pathway level 3 heatmap”. But from figure one, we can find that the composition of microbial communities in Art and Mul groups are significantly different. It is an interesting phenomenon, so I think it is better to have some discussion words in the discussion part.

Responses to Reviewers' Comments

For Reviewer #1

Comments:

Most questions I concerned before have been solved in the current manuscript. But I still have one question, in the sentence 114-115, we can find that "There was no significant difference between the Art and Mul groups in the KEGG pathway level 3 heatmap". But from figure one, we can find that the composition of microbial communities in Art and Mul groups are significantly different. It is an interesting phenomenon, so I think it is better to have some discussion words in the discussion part.

Response: Thank you very much for your suggestion. We have revised in the **related** part.

From the perspective of the species composition of the two groups, their dominant genera were the same, and the differences were mainly reflected in the community structure, as follows (**Fig. 1**): 1. The relative proportion of dominant genera were different; 2. there were a large number of low-abundance genera in the Mul group (Others: 41.16%, combined with less than 0.01% of genera).

In the case of the same dominant genera, it is acceptable that there was no significant difference in the top 20 KEGG pathways (pathway level 3), and from the **Table S4** we can see that many differences exist in the lower abundance of KO (Too trivial and messy, so there was no in-depth analysis in this manuscript).

We have also used BugBase analysis to supplement it, but since functional predictions of the amplified region of 799F–1193R alone are still limited, subsequent metabolome analysis of the intestinal fluid may make this part clearer, and this work we are working on.

October 7, 2022

Prof. Qingyou Xia
Biological Science Research Center of Southwest University; State Key Laboratory of Silkworm Genome Biology
State Key Laboratory of Silkworm Genome Biology, Southwest University
Chongqing 400715
China

Re: Spectrum02357-22R2 (Effects of Food-Gut-Feces Microbial Circulation Transfer on the Health of *Bombyx mori*)

Dear Prof. Qingyou Xia:

The authors have answered the concerns. It is comprehensive and interesting work.

Your manuscript has been accepted, and I am forwarding it to the ASM Journals Department for publication. You will be notified when your proofs are ready to be viewed.

Sincerely,

Chengshu Wang
Editor, Microbiology Spectrum

Journals Department
Supplemental Material: Accept
Supplemental Material: Accept
Supplemental Material FOR Publication: Accept
Supplemental Material: Accept
Supplemental Material: Accept